# Deep Extrapolation for Attribute-Enhanced Generation

**Alvin Chan***
Salesforce Research, NTU

**Ali Madani***
Salesforce Research

**Ben Krause**
Salesforce Research

**Nikhil Naik**
Salesforce Research

## Abstract

Attribute extrapolation in sample generation is challenging for deep neural networks operating beyond the training distribution. We formulate a new task for extrapolation in sequence generation, focusing on natural language and proteins, and propose GENhance, a generative framework that enhances attributes through a learned latent space. Trained on movie reviews and a computed protein stability dataset, GENhance can generate strongly-positive text reviews and highly stable protein sequences without being exposed to similar data during training. We release our benchmark tasks and models to contribute to the study of generative modeling extrapolation and data-driven design in biology and chemistry: https://github.com/salesforce/genhance. []

## 1 Introduction

Deep generative neural networks can generate realistic data across data-types, from sequences to images to time-series data, with applications in domains such as natural language processing (NLP), computer vision, and speech. Beyond canonical domains, the scientific application of synthetic design of proteins, molecules, and materials can be cast as generative modeling of sequences, graphs, or images (Anand & Huang, 2018; De Cao & Kipf, 2018; Madani et al., 2020, 2021). Most often, the goal is to design or generate a sample that improves upon the attribute label of interest (Fig. 1-(left)), which we term attribute-enhanced generation. Examples include generating a protein sequence with higher binding affinity or a nanomaterial structure with an energetically favorable state, as compared to all of the samples in the training distribution. In these scientific fields, traditional methods for synthetic object design with improved attributes are iterative and expensive, relying on labor- or compute-intensive methods (Bepler & Berger, 2021; Wu et al., 2021; Hie & Yang, 2021). Hence, deep generative models that can design new proteins, molecules, and materials with improved attributes have the potential to dramatically accelerate design research. Beyond scientific applications, extrapolation in generation has potential applications in NLP, such as reducing toxicity or operating in low-resource settings.

It is, however, a well-known challenge for deep neural networks to generate samples beyond the training distribution (Arora et al., 2017; Radford et al., 2019; Xu et al., 2020). In this work, we develop a method for extrapolation, particularly for sequences. Our approach, called GENhance, is designed to generate an enhanced sequence using a learned latent space. GENhance consists of a generator (sampler) and a discriminator (ranker) that are jointly trained to minimize generation and discrimination losses, regularized by latent vector smoothing and a cycle-consistency loss.

We evaluate GENhance in two data domains. First, we use the Stanford Sentiment Treebank (SST), a natural language benchmark containing movie reviews with five discrete sentiment attributes (Socher et al., 2013), to show that GENhance generates strongly positive reviews, after training with no

---

* Equal Contribution
Correspondence to amadani@salesforce.com

35th Conference on Neural Information Processing Systems (NeurIPS 2021).

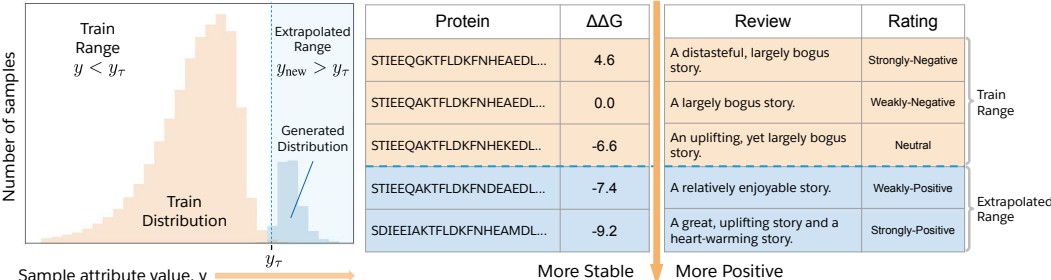

Figure 1: Attribute-enhanced generation. The goal of extrapolation (left) is to generate samples whose attribute values exceed that of all training samples, $y_\tau$. We explore target attribute extrapolation for protein sequences (center) and movie reviews (right), where more stable protein sequences and more positives text reviews are generated.

positive examples. Second, we develop a protein stability dataset for the ACE2 protein (Chan et al., 2020) with a change in free energy (ddG) continuous attribute, and show that GENhance can generate protein sequences with higher stability than the training set (Fig. 1-(right)). GENhance significantly outperforms baseline methods based on (i) a generator-discriminator model with rejection sampling and (ii) an algorithm using Metropolis-Hastings Markov chain Monte Carlo sampling with a trained discriminator. GENhance's performance is further improved when provided access to a few examples with attribute scores beyond the training distribution. Our contributions are summarized below:

- We formalize the task of extrapolation for deep generative models, focused on enhancing attributes in sequence generation, with important scientific applications in synthetic object design.

- We introduce GENhance, a regularized encoder-decoder framework with a learned latent space, and demonstrate its superior performance with respect to rigorous baseline techniques.

- We curate extrapolation benchmarks in NLP and proteins. We release the data, evaluation metrics, along with oracle models and scripts for automatic evaluation of generation quality.

## 2   Related Work

**Generalization to Low Data Regimes:**   Previous approaches aim to generalize classification and regression to low data settings. Imbalanced classification methods upsample or downsample classes(Chawla et al., 2002; García & Herrera, 2009) or reweight the training cost function (Huang et al., 2016; Cao et al., 2019; Cui et al., 2019). Yang et al. (2021) improve the generalization of regression models in extrapolation and interpolation of the continuous data domain by smoothing both the label and features of the training data. Unlike prior work in this area, GENhance aims to generate samples in low/no data settings. Methods that can better generalize discriminators to these regions are complimentary and orthogonal to our work.

**Data-Driven Design:**   Data-driven design aims to learn a distribution over a high-dimensional input space that is optimized for a fitness function corresponding to a desirable property. Design methods often iterate sampling from a generator, and then updating the generator to assign a higher probability to inputs that a discriminator predicts to have higher fitness (Bedbrook et al., 2019; Biswas et al., 2021; Mansouri Tehrani et al., 2018). Auto-focused oracles (Fannjiang & Listgarten, 2020) also adapt discriminators throughout this optimization process to using re-weighting of the training examples in the cost function to make them more reliable in the regions where the generator is more likely to generate. CbAS (Brookes et al., 2019) and DbAS (Brookes & Listgarten, 2018) use a fixed discriminator/oracle model and iteratively learns the distribution of inputs conditioned on a desirable property using importance sampling. CbAS is an improved version of DbAS which also re-weights samples based on how close they are to the original training data. We view these techniques as complementary as GENhance proposes a model-specific architecture for optimizing attributes.

Das et al. (2021) train VAE to learn latent space and use latent space classifiers to sample latent vectors through rejection sampling and decode them into sequences that would have the target attribute/label. Hawkins-Hooker et al. (2021) also decode generations of a VAE by conditioning on latent vectors

that correspond to the target attribute/label. Hoffman et al. (2020) seek to optimize molecular designs by using zeroth-order optimization on query-based prediction of candidate molecules' properties. Gómez-Bombarelli et al. (2018) build a Gaussian Process (GP) regression model trained with latent vectors to predict their inputs' labels and use gradient-based optimization on the GP to find sequences with target attributes. Compared with these previous works, the core difference in our approach is the combination of cycle-consistency and contrastive discriminatory objective to train the generator and discriminator as one model.

**Controllable Text Generation:** Our work is also related to controllable text generation, which aims to generate text that corresponds to a user-specified attribute (Kikuchi et al., 2016; Ficler & Goldberg, 2017). CTRL (Keskar et al., 2019) generates controlled fluent texts through the use of control codes which are meta-data prepended to the text during generation. Krause et al. (2020) use a generative discriminator resulting from contrasting predictions from opposing control codes to guide generation. CoCon (Chan et al., 2021) performs zero-shot controllable text generation without attribute labels. (Ziegler et al., 2019) optimizes language generation for desirable attributes via human in-the-loop reinforcement learning. Similarly to GENhance, PPLM (Dathathri et al., 2020) applies a discriminator on top of the latent of a generative model to guide generation, however, GENhance uses an autoencoder rather than a language model. Lastly, text style transfer methods have used autoencoders with disentangled style latent representations (Shen et al., 2017; Hu et al., 2017; Yang et al., 2018). Unlike text style transfer and previous approaches toward controllable text generation, GENhance differs, aside from its model formulation, in that the goal is to optimize and extrapolate a particular attribute beyond the training distribution.

# 3 Methods

Our goal is to generate sequences with target attribute values that are better than the the training data. Formally, assume that there is a ground-truth oracle ($O$) that maps each sample ($\mathbf{x} \in \mathbb{R}^d$) to the target attribute value ($y \in \mathbb{R}$) , i.e., $y = O(\mathbf{x})$. Given a dataset of oracle labeled samples ($D$), we aim to generate new sequences where its ground-truth attribute value is better than that of this dataset:

$$y_{\text{new}} > y_\tau, \quad \forall (\mathbf{x}, y) \in D : y < y_\tau \tag{1}$$

To generate samples that satisfy this criterion with high probability, we develop a sampling-ranking framework that consists of a *sampler $S$* that proposes a pool of candidate sequences and a *ranker $R$* model to infer the relative scores of these candidates. First, we describe two baseline generation techniques that are natural choices for this task and build on them to develop our GENhance model.

## 3.1 Generator-Discriminator Rejection Sampling

The first baseline, Gen-Disc, is a rejection sampling approach that uses a generator model as the sampler $S$ and a separate discriminator model as the ranker $R$ . The generator is trained to model the training distribution $p(\mathbf{x})$ through a language modeling objective where it learns to auto-regressively (Manning et al., 1999; Bengio et al., 2003) construct the training samples,

$$p(x_t, \ldots, x_l | x_1, \ldots, x_{t-1}) = \prod_{i=t}^{l} p(x_i | x_1, \ldots, x_{i-1}), \quad \mathbf{x} = x_i, \ldots, x_l \tag{2}$$

where $l$ is the length of the training sequence.

The discriminator model is trained with an objective to predict the relative ranking of sequences from a pair of samples, based on their attribute. Given two training samples $(\mathbf{x}_a, y_a)$ and $(\mathbf{x}_b, y_b)$, the pairwise contrastive loss is:

$$\mathcal{L}_{\text{contrast}} = -\log \left[ \frac{1}{1 + \exp(\tilde{y}_a - \tilde{y}_b)} \right], \quad \tilde{y}_a = f_{\text{disc}}(\mathbf{x}_a), \quad y_a > y_b \tag{3}$$

where $f_{\text{disc}}$ denotes the discriminator which outputs a scalar score value for each input sequence. We employ the contrastive loss term for this objective since it can be applied to both continuous- and discrete-labeled samples. After training, we sample candidate sequences from the generator model in an auto-regressive fashion and use the discriminator to rank the sequences according to the discriminator's output score values.

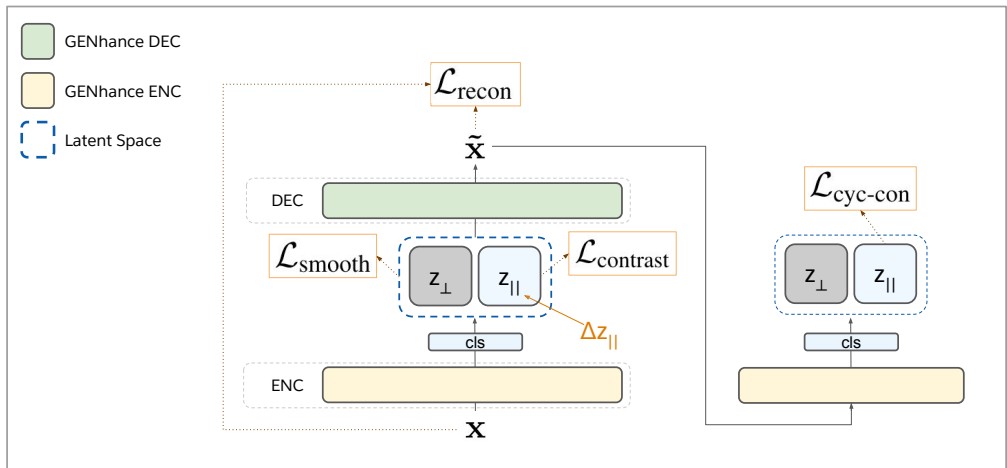

Figure 2: GENhance is an encoder-decoder framework with a latent space between the two. GENhance is trained to extrapolate beyond the training distribution of attributes, by learning the latent space using a combination of contrastive, smoothing and cycle consistency losses, in addition to the reconstruction loss for autoregressive generation.

## 3.2 Metropolis-Hastings Markov Chain Monte Carlo

Traditional methods for data-driven design rely on iterative optimization of candidates with better attributes. To mimic this process, we design a method that generates candidates using Metropolis-Hastings MCMC sampling from a population of better candidates. We start with an initial population of sequences, sampled from the training set. In the sampling step, new candidates are proposed by making edits to samples from this initial population, scored with the ranker $R$, and compared with the previous population of candidates. The probability that new generations are kept in the population depends on the score predicted by $R$. The cycle of sampling and ranking repeats until terminated. The ranker $R$ takes the form of a neural network, identical to the discriminator model in the Gen-Disc setup.

## 3.3 GENhance

In the Gen-Disc framework, since the generator is trained only to model the training distribution, there is no direct way to steer the generation distribution towards a certain direction beyond the training data. In the MCMC framework, it might be challenging to find desirable candidates with stochastic mutation operations, since the search space can be large and high-dimensional for many design problems (Kumar & Levine, 2019). To overcome these limitations, we propose using a learned latent space (Kingma & Welling, 2013) to control the attributes of generated sequences.

**Architecture:**    GENhance is an encoder-decoder framework with a latent space between its encoder ($ENC$) and decoder ($DEC$) modules (Figure 2). The latent vector ($\mathbf{z} \in \mathbb{R}^{d_z}$) of a particular input sequence ($\mathbf{x} \in \mathbb{R}^{d_x}$) is the output from the encoder module, i.e., $\mathbf{z} = ENC(\mathbf{x})$. In our experiments, $\mathbf{z}$ is the hidden state at the location of a $<cls>$ token (Devlin et al., 2018) that is prepended to the input sequence. Within the latent vector $\mathbf{z}$, representation relevant and irrelevant to the attribute of interest is stored in $\mathbf{z}_{||}$ and $\mathbf{z}_{\perp}$ respectively, i.e., $\mathbf{z} = [\mathbf{z}_{||}; \mathbf{z}_{\perp}]$. To train the encoder to store information about the target attribute in $\mathbf{z}_{||}$, we train it with a contrastive objective that aims to learn which of the two samples has the better value of the attribute:

$$\mathcal{L}_{\text{contrast}} = -\log\left[\frac{1}{1 + \exp(\tilde{y}_a - \tilde{y}_b)}\right], \quad \tilde{y}_a = f_{||}(\mathbf{z}_{a||}), \quad [\mathbf{z}_{a||}; \mathbf{z}_{a\perp}] = ENC(\mathbf{x}_a), \quad y_a > y_b$$

$$(4)$$

where $(\mathbf{x}_a, y_a)$ and $(\mathbf{x}_b, y_b)$ are a pair of training samples, each containing an input sequence $\mathbf{x}$ and its label $y$. $f_{||}$ is an operation that maps $\mathbf{z}_{||}$ to a scalar value. Here we use $\mathbf{z}_{||}$ of dimension 1 and $f_{||}$ is simply an identity operation.

We train GENhance to generate sequences using an objective where the decoder will autoregressively reconstruct a sequence while conditioned on the latent vector $\mathbf{z}$. For an input sequence $\mathbf{x}$ of length $l$, parameterizing the $ENC$ with $\theta$ and $DEC$ with $\psi$, we get the reconstruction loss:

$$\mathcal{L}_{\text{recon}} = -\sum_{i=t}^{l} log p_{\psi}(x_i|\mathbf{z}, \{x_1, \ldots, x_{i-1}\}) \quad = -\sum_{i=t}^{l} \log p_{\theta,\psi}(x_i|\mathbf{x}) \tag{5}$$

To ensure that the perturbed latent vector $\mathbf{z}$ would result in plausible generations, we include a smoothing objective, the deterministic Wasserstein autoencoder-maximum mean discrepancy (WAE-MMD) objective (Tolstikhin et al., 2017), to train the latent space as it has shown to be effective for discrete sequences. The WAE-MMD term (defined in the Supplement A.1) penalizes divergence of the latent vectors $\mathbf{z}$ from a target prior distribution $P_{\mathbf{z}}$, which is a unit Gaussian in our case.

$$\mathcal{L}_{\text{smooth}} = \text{MMD}(P_{\mathbf{z}}, \mathbf{z}) \tag{6}$$

To help learn a better latent space and stronger discriminator within GENhance, we propose a cycle-consistency learning objective ($\mathcal{L}_{\text{cyc-con}}$) to train $ENC$ to correctly predict the relative rank between two reconstructed inputs:

$$\mathcal{L}_{\text{cyc-con}} = -\log \left[ \frac{1}{1 + \exp(\hat{y}_a - \hat{y}_b)} \right], \quad \hat{y}_a = f_{||}(\hat{\mathbf{z}}_{a||}),$$
$$[\hat{\mathbf{z}}_{a||}; \hat{\mathbf{z}}_{a\perp}] = ENC(\tilde{\mathbf{x}}_a), \quad \tilde{\mathbf{x}}_a = DEC(ENC(\mathbf{x}_a)), \quad y_a > y_b \tag{7}$$

The intuition behind this objective is two-fold. First, since the discriminator ($ENC$) is used to rank generated sequences during inference, we can improve its performance on these synthetic sequences by also training the discriminator ($ENC$) on generated sequences ($\tilde{\mathbf{x}}$) during the training phase. Secondly, by backpropagating the $\mathcal{L}_{\text{cyc-con}}$ term through GENhance, it could learn a latent space which generates sequences that are easy for the discriminator to rank accurately. Combining all the training objectives, we can optimize using stochastic gradient descent to approximate the optimal parameters for GENhance:

$$\theta^*, \psi^* = \arg\min_{\theta,\psi}(\lambda_{\text{contrast}}\mathcal{L}_{\text{contrast}} + \lambda_{\text{recon}}\mathcal{L}_{\text{recon}} + \lambda_{\text{smooth}}\mathcal{L}_{\text{smooth}} + \lambda_{\text{cyc-con}}\mathcal{L}_{\text{cyc-con}}) \tag{8}$$

To the best of our knowledge, this is the first instance of using cycle-consistency with contrastive loss to train a generative model.

**Sampling & Ranking:** After training, we can sample candidates from GENhance's latent space and rank the generated samples with the scalar scores output by the GENhance's $ENC$. First, we encode a training sample with $ENC$ to sample a latent vector $\mathbf{z}$. To obtain the latent vector encoding for a new candidate sequence with an improved attribute, we can make a perturbation ($\Delta\mathbf{z}_{||}$) to the target attribute-aligned latent component $\mathbf{z}_{||}$. At the final step, GENhance's $DEC$ conditions on this perturbed latent vector ($\mathbf{z}'$) to generate the improved candidate $\tilde{\mathbf{x}}'$:

$$\tilde{\mathbf{x}}' = DEC(\mathbf{z}'), \quad \mathbf{z}' = \left[ (\mathbf{z}_{||} + \Delta\mathbf{z}_{||}) ; \mathbf{z}_{\perp} \right], \quad [\mathbf{z}_{||}; \mathbf{z}_{\perp}] = ENC(\mathbf{x}) \tag{9}$$

The perturbation $\Delta\mathbf{z}_{||}$ is determined as the direction that increases the $f_{||}$'s score output, i.e., $\frac{\partial f_{||}(\mathbf{z}_{||})}{\partial \mathbf{z}_{||}}$. For a linear layer $f_{||}$, this term is the weight of the layer while for our case where $f_{||}$ is an identity operator, $\Delta\mathbf{z}_{||}$ is simply a scalar.

After generating a pool of candidates with GENhance, we can rank and filter out top-scoring candidates with the GENhance $ENC$'s predicted score:

$$\hat{y} = f_{||}(ENC(\tilde{\mathbf{x}}')) \tag{10}$$

## 4 Experiments and Results

### 4.1 Experiments in Natural Language with SST-5

**Dataset** The Stanford Sentiment Treebank-5 (SST-5) (Socher et al., 2013) contains movie reviews from Rotten Tomatoes, which are labeled with one of the five ordinally increasing sentiment labels:

Table 1: GENhance generates a large fraction of attribute-enhanced sequences for SST-5, when 200 'Weak-Positive' samples are present in the training set ($< 4\%$ of the training set). Metrics are computed for top-1000 ranked sequences. SST-5 test samples have a mean perplexity value of 101.3. Smoothing = Latent Smoothing, CC = Cycle-consistency.

| Model | % Positive (↑ better) | % Strong-Positive (↑ better) | Perplexity (↓ better) | $\mathbb{E}[\%SP]$ (↑ better) |
|---|---|---|---|---|
| Baseline Gen-Disc | 90.6 | 26.7 | **63.9** | 14.2 |
| MCMC-Random | 17.6 | 0.5 | 49696 | 0.17 |
| MCMC-T5 | 54.8 | 10.8 | 224 | 4.58 |
| GENhance w/o Smoothing & CC | 88.2 | 21.5 | 125 | 15.44 |
| GENhance w/o CC | 91.3 | 23.6 | 101 | 16.62 |
| GENhance | **98.7** | **49.7** | 90.5 | **44.33** |

'Strong-Negative', 'Negative', 'Neutral', 'Positive', and 'Strong-Positive'. This allows us to study enhancement approaches for discrete ground-truth labels. We curate two data splits. The first SST-5 *200-Pos* setup removes all 'Strong-Positive' examples from the training set while keeping 200 randomly sampled 'Weak-Positive' examples, to simulate the presence of a small amount of higher attribute samples. For the more challenging SST-5 *No-Pos* setup, both 'Weak-Positive' and 'Strong-Positive' samples are removed. The 200-Pos and No-Pos training set have 5134 and 4934 samples respectively.

**Training** For both the Gen-Disc and MCMC models, we train the discriminator model by finetuning a publicly available (Wolf et al., 2019) pretrained T5-base encoder (Raffel et al., 2019). The generator module of both Gen-Disc and GENhance are trained by finetuning the whole pretrained T5-base encoder-decoder model. The Gen-Disc generator is trained with a language modeling objective by feeding in an empty string as the encoder's input and minimizing the cross-entropy loss between the decoder's output tokens and training sample's tokens through teacher forcing. For GENhance, the training samples are both fed in as the T5 encoder's input and used as the label for the decoder's output for the reconstruction objective. Further details on training settings on four NVIDIA A100 GPUs are found in the Supplement A.1 to A.3.

**Evaluation:** We generate 25,000 candidate sequences from each model and use their respective discriminator module to rank the sequences into pools of top-100, 1000 and 10000 sequences. The percentage of candidates containing target attributes ('Strong-Positive' & 'Weak-Positive') are computed by using a ground-truth oracle model. In our experiments, we use a pretrained BERT-large (Devlin et al., 2018) model that is finetuned on the full training SST-5 training set (including 'Strong-Positive' & 'Weak-Positive' samples), with a classification objective. This oracle model is trained with a batch size of 32 for 30 epochs and achieves an accuracy of 92.5% for strong-positive vs neutral/negative classification. 'Neutral'-labeled SST-5 sequences are used as the initial sequences for the MCMC baselines and as the input sequence for the GENhance models. $\Delta z_{||}$ perturbations of magnitude equal to 5% of the standard deviation of the training samples' $z_{||}$ are used for all GENhance generations.

We develop an additional performance metric, $\mathbb{E}[\%SP]$, the expected percentage of 'Strong-Positive' generated sequences. The metric was developed to (i) have a statistically-relevant measure with an expectation value and (ii) use the 'Strong-Positive' labels alone to maximize Oracle label fidelity, as 'Strong-Positive' labels are almost perfectly distinguishable from the train labels of 'Neutral' and lower. It is computed with the following steps: a) Randomly sample 1000 of generations from the 25000 generations, b) filter out top-100 candidates based on discriminator's ranking, c) compute % 'Strong-Positive' in top-100 with ground-truth oracle model and d) repeat step a) to c) for 100 rounds and average % strong-positive.

As a proxy for text quality, we compute the perplexity value for each generation using a pretrained GPT-2 large model (Radford et al., 2019) and average their values across the top-K pools. To guide the MCMC substitution step (MCMC-T5), we use a pretrained T5-base model, since random token substitution would degrade fluency. During mutation, a span of 1 or 2 tokens is masked and the masked sequence is fed into the T5 model to generate a replacement.

Finally, to further evaluate generated text aside from the oracle model's scores, we conducted a human evaluation study to examine the positiveness and fluency of our text generations. The study

Table 2: GENhance generates a large fraction of attribute-enhanced sequences for SST-5, when no positive samples are present in the training set. Metrics are computed for top-1000 ranked sequences. SST-5 test samples have a mean perplexity value of 101.3. Smoothing = Latent Smoothing, CC = Cycle-consistency.

| Model | % Positive (↑ better) | % Strong-Positive (↑ better) | Perplexity (↓ better) | $\mathbb{E}[\%SP]$ (↑ better) |
|---|---|---|---|---|
| Baseline Gen-Disc | 65.1 | 11.4 | **61.7** | 7.65 |
| MCMC-Random | 22.9 | 0.3 | 20924 | 0.28 |
| MCMC-T5 | 46.4 | 6.2 | 125 | 5.81 |
| GENhance w/o Smoothing & CC | 42.3 | 5.6 | 596 | 5.46 |
| GENhance w/o CC | 69.5 | 9.3 | 126 | 7.8 |
| GENhance | **87.7** | **21.4** | 118 | **19.52** |

was formulated as an A/B test where three evaluators were asked to compare pairs of text in a blinded random manner to separately determine which text was more positive or more fluent than the other. The comparisons were between text generated by GENhance vs 1) the Gen-Disc baseline, 2) MCMC baseline, 3) SST5 train data labeled as neutral, and 4) SST5 train data labeled as positive. The evaluators compared 100 samples for each of the four comparisons in the 200-Pos and No-Pos settings, totaling to 800 A/B test comparisons. For each comparison, the majority answer between the three evaluators was assigned as the final score.

**Results:** GENhance outperforms all baselines and ablation variants for all % positive metrics (Table 1 and 2). 49.7% of the more challenging 'Strong-Positive' sequences generated by GENhance are correct, which is almost twice the % of samples generated by Gen-Disc, the strongest baseline. All models see performance drops in the % positive metrics for the No-Pos training setup as compared to the 200-Pos setup, except for MCMC-Random, which is significantly worse in both cases. This reflects the greater challenge in generating desirable candidates when there are no positive samples in the training data. GENhance also outperforms other baselines in the top-1000 and top-10000 pools of candidates (See Supplement A.5). Figure 3 shows that the baselines and GENhance can generate more positive sequences than the training, with GENhance showing the largest distribution shift towards candidates with enhanced positiveness.

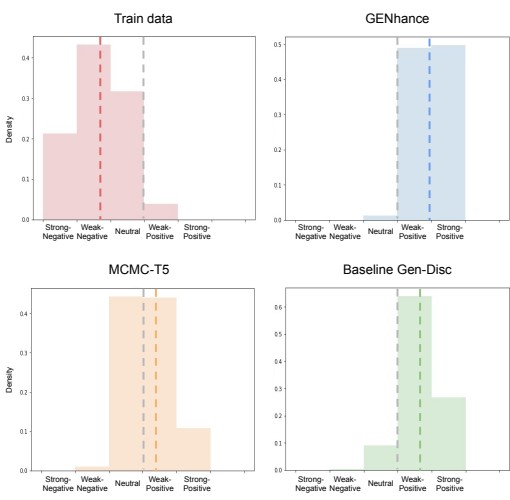

Figure 3: GENhance generations have a higher proportion of 'Strong-Positive' samples than the two baselines, and successfully extrapolate from the training data distribution. Shown here are top-1000 ranked generations for 200-Pos. Colored vertical lines show the generations' mean value.

GENhance generations also have lower perplexity values (i.e., better text quality) than the baseline and ablation methods, except for Gen-Disc, which explicitly models the training distribution (more in Supplement A.5). In fact, the average GENhance perplexity value (118) is close to the perplexity of SST-5 test samples (101.3).

According to the human evaluation (Table 4), GENhance succeeds in generating positive, fluent text that outperforms baselines. In the setup where the models were exposed to 200 positive training samples (200-Pos), GENhance outperformed all baselines, including 'Neutral' and 'Weak-Positive' training samples, in the positiveness of text generations. For the setting where the models were not exposed to any positive samples (No-Pos), GENhance is comparable to the positive train samples in

Table 3: GENhance enhances 'neutral' sequences from SST-5 to be 'strongly-positive', as seen in these generated samples. Positive words in blue and negative words in red.

| Original Text (Attribute: 'Neutral') | Generated Text (Attribute: 'Strongly-positive') |
|---|---|
| A melancholy, emotional film. | A melodramatic film, this is a powerful story. |
| An ambitious and moving but bleak film. | An ambitious and light-hearted film, it is a strong and moving story. |
| Some stunning visuals – and some staggeringly boring cinema. | An engaging collection of fantastic visuals – and yet at the same time stunningly striking. |
| You'll laugh for not quite and hour and a half, but come out feeling strangely unsatisfied. | You will laugh very well and laugh so much you will end up feeling great afterwards |
| A dark, dull thriller with a parting shot that misfires. | A dark and compelling thriller that ends with a bitter, compelling punch. |

positiveness and outperforms all the other baselines. Likewise, the fluency of GENhance's generations either match or outperform that of the training samples and baselines.

Table 4: In a human evaluation experiment of text generations on positiveness and fluency, GENhance outperforms baseline methods in both 'positiveness' and 'fluency' in most cases. Values reported in % (↑ better for all metrics).

| Model | 200-Pos | | No-Pos | |
|---|---|---|---|---|
| | Positiveness | Fluency | Positiveness | Fluency |
| Train Neutral | 7 | 30 | 18 | 39 |
| GENhance | **91** | **52** | **70** | **49** |
| Train Weak-Positive | 26 | 39 | **48** | **50** |
| GENhance | **63** | 39 | 45 | 34 |
| Gen-Disc | 29 | **42** | 35 | **44** |
| GENhance | **68** | 41 | **53** | 43 |
| MCMC-T5 | 17 | 18 | 34 | **46** |
| GENhance | **76** | **61** | **56** | 36 |

**Ablation Study:** Both latent smoothing and cycle-consistency objectives contribute to generating sequences with improved attributes and text quality. Without the cycle-consistency objective, we observe a drop in performance across all metrics, indicating that the objective is vital in helping the latent space and encoder generalize to sequences outside the training distribution. When the latent smoothing objective is removed, especially for the more challenging No-Pos setup, the generation quality drops, as indicated by the large increase in perplexity. This indicates that the smoothing objective is important in learning a latent space that is amenable to perturbations that control its attributes while maintaining generation quality.

## 4.2 Experiments in Protein Design with ACE2 Proteins

**Dataset:** Designing a protein with an optimized property (e.g. stability) is of immense interest to synthetic biology and drug discovery. Here, we create a new synthetic dataset of stability for mutations of human angiotensin-converting enzyme 2 (ACE2) protein. Since the SARS-CoV-2 virus binds to ACE2 to gain entry into human organs, ACE2 has emerged as a promising target for COVID-19 therapeutic protein design (Chan et al., 2020). Our optimization problem is to generate an ACE2-like protein sequence that is more stable than samples in the training set. As a proxy for experimentally measured stability of a protein sequence, we use the free energy calculation via FoldX (Schymkowitz et al., 2005) which provides an automated, computational oracle for testing extrapolation methods *in silico*. In particular, we measure the change in free energy from wild-type, ddG or $\Delta\Delta G$, between the folded and unfolded state of a protein sequence with the known ACE2 structure. A lower ddG value indicates a sequence that is more stable.

We mutate the N-terminus subdomain of ACE2. The protein is represented as a sequence of 83 amino acids starting from the N-terminus side, with a vocabulary of 20 amino acids. We curate 250K ACE2 variants by mutating the wild-type (natural) ACE2 subdomain through substitutions and

Table 5: GENhance generates a large fraction of highly stable ACE2-like sequences, with better mean stability as compared to baselines. Metrics are computed for top-100 ranked sequences. Smoothing = Latent Smoothing, CC = Cycle-consistency.

| Model | $ddG$ mean ($\downarrow$ better) | PCI$_{y_\tau}$ (%) ($\uparrow$ better) | $\mathbb{E}[\min]$ ($\downarrow$ better) |
|---|---|---|---|
| Baseline Gen-Disc | -4.05 | 3 | -6.18 |
| MCMC | -4.84 | 9 | -6.89 |
| DbAS | -3.48 | 2 | -4.67 |
| CbAS | -4.31 | 5 | -6.49 |
| GENhance w/o Smoothing & CC | -7.16 | 66 | -8.31 |
| GENhance w/o CC | -5.59 | 30 | -7.68 |
| GENhance | **-7.34** | **77** | **-8.71** |

computing their ddG values. To keep a local landscape of deviation from wild-type, amino acids were substituted by another amino acid with a probability of $P = 4/L$ where $L$ is the length of the protein's mutable region. Mutants with more than eight mutations are discarded and a constant region (NTNITEEN) is maintained. The ddG values for each sequence are computed as the average over five FoldX simulations. More details are in Supplement A.4. We use this dataset to evaluate GENhance's ability to generate protein sequences with lower ddG values than those found in the training distribution. In contrast to SST-5, the ACE2 ddG values lie on a continuous scale, allowing us to validate GENhance in the continuous label setting.

**Training:** To initialize the weights of our models, we use a T5-base model that is pretrained on Uniref50 (Suzek et al., 2015) with a masked span objective of mean length 3. The discriminator model for both Gen-Disc and MCMC models are trained by finetuning the pretrained encoder on the full set of 250K sequences, with a random 10% used as the validation set while the generator modules of both Gen-Disc and GENhance are trained by finetuning the whole pretrained encoder-decoder model. Further details on training settings on four NVIDIA A100 GPUs are found in the Supplement.

**Evaluation:** For evaluation, we generate 250,000 sequences from each model while eliminating generations without the constant region (NTNITEEN) or with a different sequence length from the wild-type sequence. We then use the methods' respective discriminator modules to rank the candidates into pools of top-10, 100 and 1000 sequences. The top-K sequences' ddG values are then computed with the FoldX software, taking the average over five simulation runs. The top-5% most stable sequences are used as the initial sequences for MCMC and as the input sequence for GENhance. $\Delta z_{||}$ perturbations of magnitude equal to $25\%$ of the std. dev. of the training samples' $z_{||}$ are used for all the GENhance models. Following Fannjiang & Listgarten (2020), we also measure percent chance of improvement (PCI$_{y_\tau}$) over the best label (most negative ddG) in training data. To have a statistically-relevant metric, we developed the expected minimum ddG value ($\mathbb{E}[\min]$) which is computed by the following steps: a) Randomly sample 10000 of generations from the 250,000 generations, b) filter out top-10 candidates based on discriminator's ranking, c) compute ddG top-

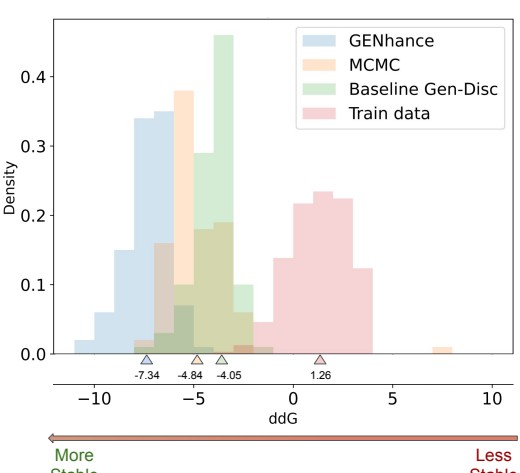

Figure 4: GENhance-generated ACE2-like sequences show the largest shift in ddG distribution (i.e., stability improvement) from the training set. Top-100 ranked generated sequences are shown. ddG values are binned with an interval of 1. Triangles denote the distributions' mean values.

10 with FoldX oracle software to find the minimum ddG value among these 10 candidates and d) repeat step a) to c) for 100 rounds and average the minimum ddG values across these rounds.

In addition to Gen-Disc and MCMC, we include CbAS (Brookes et al., 2019) and DbAS (Brookes & Listgarten, 2018) as baselines for comparison in this task. Both CbAS and DbAS use the same model architecture as the baseline Gen-Disc. We first sample generations from the baseline Gen-Disc's generator then retrain the generator on the generations re-weighted by the discriminator's scores. We used the initial hyperparameters from CbAS and conducted a grid search on a) M, number of generation per iterations (50, 100, 200), b) Q, percentile threshold (75, 90), c) temperature of a sigmoid score-based weight computation (0.1, 1, 10) and report the best results for CbAS. The DbAS' hyperparameter values mirror those of CbAS in our experiments.

**Results:** GENhance outperforms all baselines on all metrics (Table 5) in designing more stable sequences. GENhance sequences have the lowest mean ddG value, with a significant fraction of generations more stable than the most stable sequence in the training set, as indicated by the higher $PCI_{y_\tau}$ values. GENhance also has the lowest $\mathbb{E}[\min]$ value, indicating that it may be well-suited to find stable protein candidates in laboratory experiments where only small numbers of candidates can be evaluated due to cost. Even though MCMC and CbAS fare better than the simpler baseline Gen-Disc setup, we observe that GENhance outperforms both baselines on all three metrics measured. The distribution of generated samples by GENhance shows the largest shift towards more stable sequences as compared to the original training distribution (Figure 4).

**Ablation Study:** Similar to SST-5 experiments, GENhance outperforms its ablation variants on all metrics. Rather surprisingly, we observe a drop in performance when the latent smoothing objective is added, which we speculate is due to the tension between the GENhance's reconstruction and its encoder's contrastive training objectives. With the cycle-consistency objective, we see a boost to GENhance that outperforms the vanilla variant, indicating that this objective aids in stabilizing the convergence of these two training objectives. To further study their contribution to GENhance's performance, we use GENhance's encoder to rank sequences generated by the generator in the baseline Gen-Disc setup and observe a boost in performance (Supplement Table 12). This suggests that GENhance's superior performance is due to both more accurate ranking by its encoder and better generation by its decoder.

**Discussion:** There are two main features of GENhance that may contribute to its ability to generate attribute-enhanced sequences. First, compared to the baselines which mainly rely on the discriminator's prediction to filter out promising candidates, GENhance can additionally use its latent space to steer the general distribution of generated candidates towards a target region (e.g., more stable or positive sequences). Second, unlike the discriminators in the baselines which were trained only on training samples, the encoder used in GENhance was also trained on GENhance-generated sequences through the cycle-consistency loss. This may contribute to the better-ranking performance for GENhance, increasing the fraction of desirable candidates.

# 5 Conclusion

In conclusion, we formalize the task of attribute-enhanced generation that aims to create improved samples with target attributes beyond the training distribution. Scientific applications can include the design of proteins, materials, and molecules without expensive, iterative procedures for discovery. To achieve this, we proposed GENhance, a generative model with a trained latent space, that generates sequences that outperform both the training data and baseline methods in natural language and protein engineering tasks. In the future, we aim to expand GENhance to other data types beyond sequences and study generation in scenarios where new data samples could be actively acquired. We also open-source our curated benchmark datasets with computational oracles along with all models and evaluation metrics/scripts to enable further research in extrapolation: https://github.com/salesforce/genhance.

**Broader Impact:** Extrapolation involves designing samples, whether text, proteins, molecules, or materials, that have attributes that are unseen in training. If our technique or a future iteration thereof is adopted broadly, care should be taken in terms of the end use-cases of these designed/optimized samples and downstream effects to ensure safe, non-nefarious, and ethical applications. For projects in any domain, active oversight during project initiation, experimental optimization, and deployment phases should be put in place to ensure safe usage and limitation of unintended harmful effects.

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
