# A Supplementary Material

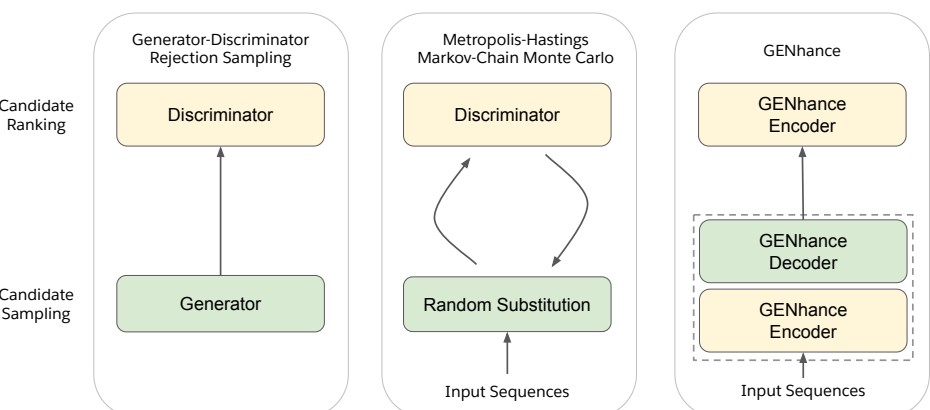

Figure 5: Comparison between baseline candidate sampling techniques and GENhance. (left) In a baseline generator-discriminator setup, a generator is trained to model the distribution of training sequences while a discriminator is trained to predict input sequences' attribute values ($\Delta\Delta G$). When sampling candidates from the generator, the discriminator model ranks generated sequences which can then be evaluated by ground-truth oracle for evaluation. (center) In MH-MCMC, instead of a generator model, candidate sequences are sampled through an iterative process where highest-ranked generations are kept in the population and randomly mutated to generate offspring candidate sequences. (right) GENhance acts as a generator that conditions on an input sequence. Instead of a separately trained discriminator, the GENhance encoder is used to rank the candidate sequences.

## A.1 GENhance - Model Training and MMD Formulation

For GENhance, the training sample are both fed in as the MT5 encoder's input and used as the label for the decoder's output for the reconstruction objective. We use only the 125K most stable sequences as the training set to bias the generation towards more stable candidates. We use a warmup phase for GENhance where the WAE-MMD and cycle-consistency objectives are turned on in the midpoint of the training phase for better training stability (Li et al., 2019). $\lambda$ is set to 1 for all secondary objectives. For both 200-Pos and No-Pos, 10% of the training samples are randomly selected to make up the validation set. All GENhance models are trained with a linear learning rate scheduler and max learning rate of 1e-4. For SST-5, all GENhance variants are trained for 25 epochs, with batch size of 16. For the ACE2 task, GENhance w/o smoothing & CC (cycle-consistency) is trained for 12 epochs while the GENhance w/o CC (cycle-consistency) and full version are trained for 24 epochs for full convergence.

**WAE-MMD Objective** The WAE-MMD objective used in our experiments largely follows the setting in Tolstikhin et al. (2017). For a positive-definite reproducing kernel $k : \mathcal{Z} \times \mathcal{Z} \rightarrow \mathbb{R}$, the maximum mean discrepancy ($MMD_k$) is:

$$\text{MMD}_k(P_{\mathbf{z}}, Q_{\mathbf{z}}) = \| \int_{\mathcal{Z}} k(\mathbf{z}, \cdot) dP_{\mathbf{z}}(\mathbf{z}) - \int_{\mathcal{Z}} k(\mathbf{z}, \cdot) dQ_{\mathbf{z}}(\mathbf{z}) \|_{\mathcal{H}_k} \quad (11)$$

where $\mathcal{H}_k$ is the RKHS of real-valued functions mapping $\mathcal{Z}$ to $\mathbb{R}$. If $k$ is characteristic then $\text{MMD}_k$ defines a metric and can be used as a divergence measure. Since MMD has an unbiased U-statistic estimator, it can be used together with stochastic gradient descent (SGD) during training in the following form:

$$\text{MMD}_k(P_{\mathbf{z}}, Q_{\mathbf{z}}) = \frac{1}{n(n-1)} \sum_{l \neq j} k(\mathbf{z}_l, \mathbf{z}_j) + \frac{1}{n(n-1)} \sum_{l \neq j} k(\tilde{\mathbf{z}}_l, \tilde{\mathbf{z}}_j) - \frac{2}{n^2} \sum_{l,j} k(\mathbf{z}_l, \tilde{\mathbf{z}}_j),$$
$$\{\mathbf{z}_1, \ldots, \mathbf{z}_n\} \sim P_{\mathbf{z}}, \quad \{\tilde{\mathbf{z}}_1, \ldots, \tilde{\mathbf{z}}_n\} \sim Q_{\mathbf{z}} \quad (12)$$

where $\{\mathbf{z}_1, \ldots, \mathbf{z}_n\} \sim P_{\mathbf{z}}$ are samples from the target prior distribution $P_{\mathbf{z}}$ and $\{\tilde{\mathbf{z}}_1, \ldots, \tilde{\mathbf{z}}_n\} \sim Q_{\mathbf{z}}$ are the decoded latent vectors in GENhance. For $k$, we use a random-feature Gaussian kernel with $\sigma = 14$ and random feature dimension of 500.

## A.2 Baseline MCMC Algorithm and Parameters

The discriminator used in the MCMC algorithm is trained with a linear learning rate scheduler and max learning rate of 1e-4. For SST-5, with a random 10% used as the validation set. For SST-5 experiments, the discriminator is trained with a learning rate of 1e-4 and a batch size of 16 for 25 epochs with the contrastive objective. For ACE2 experiments, we train the discriminator model by finetuning a pretrained protein T5-base encoder on the full set of 250K sequences, with a random 10% used as the validation set. The discriminator is trained with a learning rate of 5e-6 and a batch size of 32 for 10 epochs with the contrastive objective.

The MCMC algorithm used in our experiments largely follows the setup in Biswas et al. (2021). The steps for the MCMC algorithm are:

1. Initialize: set the initial sequence as state sequence, $s$.

2. Propose a new sequence $s^*$ by editing $s$ (e.g., token/span substitution).

3. Compute the acceptance probabililty: $\min\left[1, \exp\left(\frac{\tilde{y}^* - \tilde{y}}{T}\right)\right]$ where $\tilde{y}^*$ and $\tilde{y}$ are the fitness score of the proposed sequence $s^*$ and state sequence $s$ respectively and $T$ is the temperature. In our experiments, the fitness score is the predicted score from the discriminator model, i.e., $\tilde{y} = f_{\text{disc}}(s)$.

4. If accepted, set proposed sequence as the new state sequence: $s \leftarrow s^*$.

5. Repeat step 2 to 4 for a predefined number of iterations.

For SST-5, we use a temperature $T = 0.1$ and 100 iterations. Any sequence candidates with an Levenshtein distance larger than 30% the length of the original text sequence are not accepted in the population pool. For the ACE2 experiments, we use a temperature $T = 0.1$ and 1000 iterations. Any sequence candidates with an edit distance larger than 18 (vs the wild-type sequence) are not accepted in the population pool.

The MCMC's mutation rate, determined by the value of temperature, relates to how often the initial sequence is replaced by a fitter mutant sequence. A higher temperature will produce more explorative trajectories which may get trapped in low fitness regions while a lower temperature results in more exploitative trajectories which may not advance beyond local optima. The number of MCMC iterations determines the length of the mutation trajectory. A large number of iterations may result in sequences that are highly mutated, lying far outside the distribution of the training data, resulting in poor discriminator ranking performance. Conversely, a small number of iterations result in smaller edits that may not be enough to reach the global optima of the fitness landscape. We will include this discussion in the revision and refer readers to Biswas et al. (2021)) for more details on the MCMC method.

We conducted a grid search for the best MCMC parameters: mutation rate/temperature (4 orders of magnitude), MCMC iterations (3 orders of magnitude) and edit distance constraints (3 values) and report the best results in this paper.

## A.3 Baseline Gen-Disc Details

The discriminator used in the baseline Gen-Disc algorithm is trained with a linear learning rate scheduler and max learning rate of 1e-4. For SST-5, with a random 10% used as the validation set. For SST-5 experiments, the discriminator is trained with a learning rate of 1e-4 and a batch size of 16 for 25 epochs with the contrastive objective. For ACE2 experiments, the discriminator is trained with a learning rate of 5e-6 and a batch size of 32 for 10 epochs with the contrastive objective.

The baseline generator is trained with a language modeling objective by feeding in a empty string as the encoder's input and minimizing the CE loss between the decoder's output tokens and training sample's tokens through teacher forcing. The generator is trained with a batch size of 16, a linear learning rate scheduler and max learning rate of 1e-4. The generator is trained for 25 epochs for the

SST-5 experiments and for 12 epochs for the ACE2 experiments. The generator is trained on the full training set for the SST-5 experiments and on the top 50% most stable protein sequences for the ACE experiments.

### A.4  Ground-Truth Oracles

**SST-5**  For the ground-truth oracle model, we use a pretrained BERT-large (Devlin et al., 2018) model that is finetuned on the full training SST-5 training set (including 'Strong-Positive' & 'Weak-Positive' samples), with a classification objective. This oracle model is trained with a batch size of 32 for 30 epochs.

**ACE2**  The force field and energy calculations of FoldX [1] were used to calculate ddG values of the mutated protein sequences for the initial training data and generated sequences. Although energy calculations with any software have limitations in accuracy, FoldX provides a computational technique to evaluate generation quality in *high-throughput*– making it an attractive testbed for rapid experimentation of novel ML methods for attribute-enhanced generation in proteins.

The protein used within this study was human ACE2, PDB:6LZG. We particularly focused on a subdomain that interacts with the RBD domain of SARS-CoV-2 S1 spike– spanning residues S19 to Q102 inclusive. FoldX 5.0 was used with the RepairPDB and BuildModel modules to extract the change in free energy with respective to wild-type for the given crystallographic structure. The oracle evaluation model's output was an average ddG value over 5 runs. We refer readers to Usmanova et al. (2018) for best practices.

### A.5  Supplementary Tables

Table 6: Percentage (%) of SST-5 generations classified as 'Positive' (or 'Strong-Positive') for GENhance model ablation variants and baseline techniques in the 200-Pos training setup, with varying top-K values. (↑ better) for all metrics below.

| Model | % Positive | | | % Strong-Positive | | |
|---|---|---|---|---|---|---|
| | Top-100 | Top-1000 | Top-10000 | Top-100 | Top-1000 | Top-10000 |
| Baseline Gen-Disc | 99 | 90.6 | 29.2 | 64 | 26.7 | 4.39 |
| MCMC-Random | 29 | 17.6 | 4.93 | 1 | 0.5 | 0.08 |
| MCMC-T5 | 93 | 54.8 | 16.3 | 53 | 10.8 | 1.51 |
| GENhance w/o Smooth. & CC | 95 | 88.2 | 50.4 | 35 | 21.5 | 6.53 |
| GENhance w/o CC | 97 | 91.3 | 55.3 | 34 | 23.6 | 7.24 |
| GENhance | **100** | **98.7** | **89.6** | **52** | **49.7** | **27.6** |

Table 7: Perplexity score for for GENhance model ablation variants and baseline techniques in the 200-Pos training setup, with varying top-K values. (↓ better) for all values below.

| Model | Top-100 | Top-1000 | Top-10000 |
|---|---|---|---|
| Baseline Gen-Disc | **60.5** | **63.9** | **65.8** |
| MCMC-Random | 3994 | 49696 | 15394 |
| MCMC-T5 | 88.8 | 224 | 263 |
| GENhance w/o Smooth. & CC | 149 | 125 | 220 |
| GENhance w/o CC | 78.5 | 101 | 154 |
| GENhance | 102 | 90.5 | 118 |
| SST-5 Test Samples | | 101.3 | |

---

[1]FoldX Suite - http://foldxsuite.crg.eu/

Table 8: Percentage (%) of SST-5 generations classified as 'Positive' (or 'Strong-Positive') for GENhance model ablation variants and baseline techniques in the No-Pos training setup, with varying top-K values. (↑ better) for all metrics below.

| Model | % Positive | | | % Strong-Positive | | |
|---|---|---|---|---|---|---|
| | Top-100 | Top-1000 | Top-10000 | Top-100 | Top-1000 | Top-10000 |
| Baseline Gen-Disc | 74 | 65.1 | 23.8 | 8 | 11.4 | 3.11 |
| MCMC-Random | 49 | 22.9 | 6.75 | 1 | 0.3 | 0.05 |
| MCMC-T5 | 50 | 46.4 | 26.7 | 2 | 6.2 | 2.57 |
| GENhance w/o Smooth. & CC | 38 | 42.3 | 40.6 | 3 | 5.6 | 4.64 |
| GENhance w/o CC | 84 | 69.5 | 42.5 | 7 | 9.3 | 4.8 |
| GENhance | **92** | **87.7** | **76.3** | **17** | **21.4** | **15.3** |

Table 9: Perplexity score for for GENhance model ablation variants and baseline techniques in the No-Pos training setup, with varying top-K values. (↓ better) for all values below.

| Model | Top-100 | Top-1000 | Top-10000 |
|---|---|---|---|
| Baseline Gen-Disc | **48.6** | **61.7** | **68.4** |
| MCMC-Random | 9817 | 20924 | 19002 |
| MCMC-T5 | 104 | 125 | 265 |
| GENhance w/o Smooth. & CC | 465 | 596 | 201 |
| GENhance w/o CC | 100 | 126 | 184 |
| GENhance | 111 | 118 | 148.5 |
| SST-5 Test Samples | | 101.3 | |

Table 10: Mean ddG values of generations from baseline generators and GENhance model variants, with varying top-K values, for ACE protein.

| Model | Top-10 | Top-100 | Top-1000 |
|---|---|---|---|
| Baseline Gen-Disc | -4.37 | -4.05 | -3.47 |
| MCMC | -4.46 | -4.84 | -4.94 |
| GENhance w/o Smooth. & CC | **-7.98** | -7.16 | -5.73 |
| GENhance w/o CC | -5.51 | -5.59 | -5.13 |
| GENhance | -7.84 | **-7.34** | **-6.44** |

Table 11: PCI$_{y_\tau}$ values (%) of generations from baseline generators and GENhance model variants, with varying top-K values, for ACE protein.

| Model | Top-10 | Top-100 | Top-1000 |
|---|---|---|---|
| Baseline Gen-Disc | 0 | 3 | 2.9 |
| MCMC | 0 | 9 | 12.1 |
| GENhance w/o Smooth. & CC | **100** | 66 | 19.2 |
| GENhance w/o CC | 30 | 30 | 10.1 |
| GENhance | 90 | **77** | **38.5** |

Table 12: Performance when baseline Gen-Disc's generated are ranked by its own discriminator and by GENhance encoder module.

| Model | $ddG$ mean (↓ better) | PCI$_{y_\tau}$ (%) (↑ better) | $\mathbb{E}[min]$ (↓ better) |
|---|---|---|---|
| Baseline Disc | -4.05 | 3 | -6.18 |
| GENhance ENC | -4.25 | 7 | -6.05 |