# OpenReview forum: "Deep Extrapolation for Attribute-Enhanced Generation"
_NeurIPS.cc/2021/Conference — NeurIPS 2021 Poster_

### Official Review · Reviewer_1cpZ · 2021-07-15

**Rating:** 7
**Confidence:** 4

**Summary:**

The paper describes GenEnhance, a method for generating sequences with attributes that do not exist in the training distribution. GenEnhance is based on a discriminator / generator architecture trained with a cycle consistency loss similar to existing methods. The main innovation of GenEnhance is that the discriminator is trained with a contrastive loss for learning to rank generated samples by their attribute.
The paper is well written and combining a contrastive loss with a cycle consistency loss for controllable generation is new as far as I know. However, it is unclear how the cycle-consistency loss performs relative to regression losses and how GenEnhance performs relative to existing methods for model-based optimization in the visible and latent space.

**Limitations And Societal Impact:**

yes

**Main Review:**

# Major comments
1) I am considering the use of the contrastive loss for training the discriminator and combining it with the cycle-consistency loss as the main contribution of the paper. Using a cycle-consistency loss alone for training a generative model is not new. However, the ablation experiments do not show how cycle-consistency + contrastive loss performs relative to cycle-consistency + regression loss. How does the contrastive loss compare to traditional regression losses, e.g. mean-squared error?

2) It is unclear how GenEnhance performs relative to existing methods for optimizing sequences in both the latent space (as GenEnhance does) and visible space (similar to the MCMC and Gen-Disc baseline). I would like to see the following baselines:
* DbAs / CbAs (http://arxiv.org/abs/1810.03714, http://arxiv.org/abs/1901.10060): Train a discriminator (with regression loss and optionally also contrastive loss) on the training set and optimize it by cross-entropy optimization using a VAE. This baseline will be similar to the Gen-Disc baseline, with the difference that the generative model is updated by cross-entropy optimization.
* Gomez-Bombarelli (https://doi.org/10.1021/acscentsci.7b00572): Train a VAE and discriminator in the latent space and optimize the discriminator in the latent space to generate new sequences. This baseline will show how GenEnhance performs relative to Bayesian optimization performed in the latent space of a VAE.

3) How does the performance of the MCMC baseline depend on a) the mutation rate and b) the number of MCMC iterations (optimization steps)?

4) Which and how were hyper-parameters optimized?

5) The protein design experiment does not show if generative sequences are realistic. Please report the perplexity using a generative model (e.g. DCA or HMM model) fit on homologous sequences of the ACE2 parent sequence.

6) The ACE2 dataset includes 250K training samples. However, the number of labeled sequences that can be used as a starting point for the optimization is smaller in practice (in the order of dozens to hundreds). I would therefore like to see how GenEnhance performs on a dataset with fewer training samples. For example, you may consider the GFP or stability dataset described in the Rao et al (http://arxiv.org/abs/1906.08230). Also perform two different dataset splits as you did for SST-5.

# Minor comments
7) l144-146: Please show experimentally that the performance of the discriminator improves (as you assume) when it is also trained with generated samples.

8) l19: Books et al 2019 describe one specific method for protein design. I suggest to replace the reference by a more comprehensive review of the field, e.g.
https://www.sciencedirect.com/science/article/pii/S2405471221002039
https://www.sciencedirect.com/science/article/pii/S136759312100051X
http://arxiv.org/abs/2106.05466

9) Is the generator of the Gen-Disc baseline trained on all training sequences or only the best sequences (e.g. all sequences with y > 90% threshold)? Training it on only the best sequences will most likely increase the likelihood of generating sequences that are better than training sequences when sampling from it.

10) l190, accuracy of the oracle model: What is the AUC-ROC and AUC-PR? What is the fraction of strong-positive vs. neutral/negative reviews? Accuracy is in appropriate to quantify the performance of the oracle if class is imbalanced.

11) [optional]: I suggest combining Table 1 and Table 2 to make it easier for the reader to compare performances for the two dataset splits.

20) [optional]: Using the perplexity of a generative model for quantifying the quality of generated sequences SST-5 sentences has known limitations. I suggest strengthening the experiment by also evaluating the top sequences using human ratings (e.g. from Amazon Mechanical Turk).

**Time Spent Reviewing:**

3

---

> ### Author Response · Authors · 2021-08-10
> **Response to Reviewer 1cpZ**
>
> We thank the reviewer for the thoughtful feedback. We are encouraged that you find our ranking discriminator approach innovative and the paper well-written. We address your comments below:
>
> 1: Comparison of cycle-consistency + contrastive loss vs. cycle-consistency + regression loss (such as MSE):
> In the initial stages of our experimentation, we trained discriminator models on the contrastive loss and regression loss setups and found that contrastive loss outperforms the regression objective in ranking stable sequences that are outside the data distribution, i.e. bottom-90% stable sequences as training set, top-10% stable sequences as test set. The Spearman’s rank correlation on the test set for the discriminator with the contrastive loss is 0.62 and for that with the regression loss is 0.41 in that experiment---indicating that the contrastive loss enables more effective discriminative ability in out-of-distribution/extrapolation settings.
> We speculated that cycle-consistency + contrastive loss would be a more effective combination than the cycle-consistency + regression loss due to the more relaxed assumption on the labels for the contrastive objective. Since the reconstructed sequences may be different from the original input sequences (especially in the early phase of the training), the 2 ddG values (regression label) of a reconstructed sequence pair are mostly likely altered, breaking the cycle-consistency assumption, while the pairwise relative rank (contrastive label) is more likely to be preserved. As requested, we conducted additional experiments on the cycle-consistency+regression and the results for Top-100 ACE generations are $ddG$ mean: -5.1, $PCI_{y_{\tau}}$: 13%, $\mathbb{E}[\min]$: -7.01, indicating that cycle-consistency+contrastive loss is a better formulation as shown in Table 4. This may support the above point about the stronger synergy between the cycle-consistency and contrastive objective in GENhance.
>
> 2: Comparison with more baselines:
> As advised by the reviewer, we have conducted additional experiments on two more baselines (CbAS and DbAS). We followed the implementation recommended by the reviewer: using the same architecture as the baseline Gen-Disc for CbAS and DbAS. Specifically, we sample generations from the baseline Gen and retrain the generator on the generations re-weighted by the discriminator’s scores. The initial hyperparameters follow the values used in the CbAS paper and we conducted grid search on a) M, number of generation per iterations (50, 100, 200), b) Q, percentile threshold (75, 90), c) temperature of a sigmoid score-based weight computation (0.1, 1, 10) and report the best results for CbAS. The DbAS hyperparameter values mirror that of CbAS.
> With regards to Table 4, the results for CbAS are: $ddG$ mean=-4.31, $PCI_{y_{\tau}}$= 5%, $\mathbb{E}[\min]$=-6.49 while for DbAS are: $ddG$ mean=-3.48, $PCI_{y_{\tau}}$= 2%, $\mathbb{E}[\min]$=-4.67. Even though the CbAS fares better than the simpler baseline Gen-Disc setup, we observe that GENhance outperforms both baselines on all the 3 metrics measured.
>
> 3 & 4: “How does the performance of MCMC depend on a) mutation rate and b) number of MCMC iterations? Which and how were hyperparameters optimized?”
> The MCMC’s mutation rate, determined by the value of temperature, relates to how often the initial sequence is replaced by a fitter mutant sequence. A higher temperature will produce more explorative trajectories which may get trapped in low fitness regions while a lower temperature results in more exploitative trajectories which may not advance beyond local optima.
> The number of MCMC iterations determines the length of the mutation trajectory. A large number of iterations may result in sequences that are highly mutated, lying far outside the distribution of the training data, resulting in poor discriminator ranking performance. Conversely, a small number of iterations result in smaller edits that may not be enough to reach the global optima of the fitness landscape. We will include this discussion in the revision and refer readers to Biswas et al. (2021) for more details on the MCMC method.
> We conducted a grid search for the best MCMC parameters: mutation rate/temperature (4 orders of magnitude), MCMC iterations (3 orders of magnitude) and edit distance constraints (3 values) and report the best results in the paper.
>
> 5: About "The protein design experiment does not show if generative sequences are realistic":
> We thank the reviewer for the feedback. To address this concern, we fit an autoregressive generative language model on homologous sequences to ACE2, extracted from an HHBlits search on Uniref30. The generative model was utilized to compute average perplexities for the top 1000 sequences from each generation method. GENhance, Gen-Disc, and MCMC yield perplexities of 4.46, 4.96, and 7.28 respectively. According to this evaluation approach, GENhance generates the most realistic protein sequences.
>
> 6:  How does GenEnhance perform on a dataset with fewer training samples:
> While we agree that more work on smaller training datasets makes for great future work, we wish to point out that our SST-5 experiments study a similar setting with a much smaller dataset (around 5K train samples) and coarser labels (3 classes for the No-Pos setup). The SST-5 results show that GENhance outperforms the baselines in such a scenario.
>
> 7: “Please show experimentally that the performance of the discriminator improves (as you assume) when it is also trained with generated samples.”
> We thank the reviewer for the suggestion. To show GENhance improves the discriminator performance through its training with generated samples, we conducted an additional experiment that uses GENhance's discriminator to rank the top-100 generations from the baseline generator from the “Baseline Gen-Disc” setting. Even though GENhance’s discriminator has not seen the baseline generator’s output during training, using this discriminator boost the stability statistics of the top-100 sequences: with improved $ddG$ mean of -4.25 (vs -4.05 for Baseline Gen-Disc) and $PCI_{y_{\tau}}$ of 7% (vs 3%). We will include this result in the revision.
>
> 8: Replace Brookes et al. 2019 reference with more comprehensive references:
> We thank the reviewer for pointing us to these references. We will include them in the revised version.
>
> 9: Is the generator of the Gen-Disc baseline trained on all training sequences or only the best sequences (e.g. all sequences with y > 90% threshold)?
> The generator is trained on the full training set for the SST-5 experiments and on the top 50% most stable protein sequences for the ACE experiments. The reason for training on the full train set for SST-5 is due to the relatively small size of the data (around 5K samples) and the presence of only 3 classes for the No-Pos setup. For the ACE experiments, we trained the Gen-Disc baseline on the top 50% stable samples due to the large data size and wide spectrum of ddG values. Training it with a higher threshold of top 10% stable sequence in our initial experimentation yielded a lower-performing baseline Gen-Disc result, likely due to the smaller data size and diversity. We will include this discussion in the revision.
>
> 10 & 12:  Validity of the oracle model and strengthening the experiment by evaluating the top sequences using human ratings:
> To further evaluate generated text aside from the oracle discriminator scores, we performed a human evaluation study to examine the positiveness and fluency of our text generations. The study was formulated as an A/B test where three evaluators were asked to compare pairs of text in a blinded random manner to separately determine which text was more positive or more fluent than the other. The comparisons were between text generated by GENhance vs the Gen-Disc baseline, MCMC baseline, SST5 train data labeled as neutral, and SST5 train data labeled as positive. There were 100 samples evaluated for each of the 4 comparisons in the two task settings described in the paper-- which totals 800 A/B test comparisons. For each comparison, the majority answer between the three humans was assigned as the final score. The full results of the human evaluation study will be included in the updated manuscript. In short, GENhance succeeds in generating positive, fluent text that outperforms baselines. For the setting where the models were exposed to 200 positives, GENhance outperformed the Gen-Disc baseline by 68:29, outperformed the MCMC baseline by 76:17, outperformed neutral samples from the SST5 train set by 91:7, and outperformed positive samples from SST-5 train by 63:26. For the setting where the models weren’t exposed to any positives or strong positives, GENhance outperformed Gen-Disc by 53:35, outperformed MCMC by 56:34, outperformed neutral SST5 train samples by 70:18, and was comparable to positive SST5 train samples by 45:48. Likewise, the fluency of GENhance samples either match or outperform training samples and baselines.
>
> 11: combining Table 1 and Table 2
> We thank the reviewer for this suggestion. We will combine these tables in the updated version.
>
> We would like to thank the reviewer again for the very helpful and thoughtful feedback for us to improve the paper.

---

> > ### Comment · Reviewer_1cpZ · 2021-08-21
> > **Strong rebuttal**
> >
> > Thanks for rigorously addressing all my comments! I increased by score accordingly.

---

### Official Review · Reviewer_XZaE · 2021-07-16

**Rating:** 5
**Confidence:** 4

**Summary:**

This work introduces a framework for learning representations that can extrapolate attributes in text generation.

**Limitations And Societal Impact:**

Part of the potential negative societal impact was addressed in the social impact section.

**Main Review:**

- About contribution and novelty.
  - The idea proposed in this project is a special case of using discriminators to learn disentangled representations.
  - Another question about this work is the controllability. Particularly, for the case of boosting text attributes from "Neutral" to "Strongly-positive", as both "Strongly-positive" and "Weak-Positive" were not in the training set, how did the model learn the difference between these two types of attributes? Furthermore, how can we control the model to generate a "Strongly-positive" sentence instead of a "Weak-Positive" one?
  - Overall, I did not see any theoretical evidence that the proposed idea can handle the extrapolation problem discussed at the beginning of this paper
- About modeling
  - For smoothing the latent space, why not use word perturbation to create $\tilde{x}_{\alpha}$? What is the advantage of using this cycling strategy?
  - What is the justification of simplifying $z_{||}$ as a 1-dimensional scalar and $f_{||}$ as an identity operation? (line 160 - 161)
- About experiments
  - There is no comparison between the proposed models and some existing works on controllable text generation. I noticed that in table 1, several baseline models were reported. However, none of them is from the existing codebase, which should not be difficult to find.
  - There is a potential mismatch for the baseline Gen-Disc models. As explained in the paper, Gen-Disc was trained with an empty string as the input to the encoder (line 179), while for evaluation, "Neutral"-labeled sequences are used as inputs for generation (line 191). This mismatch may degrade the actual performance of baseline models. On the other hand, GENhance used text inputs during both training and evaluation.
  - In addition to the average of 100 rounds of sub-sampling, it would also be interesting to report the variance and also the overall performance on the entire 25K generated texts.
  - Recall the goal of this work is generating "sequences with target attribute values that are better than the the data" (there is an extra "the"). To verify this claim, at least for text generation, we should know the human evaluation performance.

**Time Spent Reviewing:**

2.0

---

> ### Author Response · Authors · 2021-08-10
> **Response to Reviewer XzaE**
>
> We thank the reviewer for the thoughtful and constructive comments. Please refer to the following for our response:
>
> - Regarding the contribution and novelty:
> Apart from using a discriminator to learn disentangled representations through a contrastive loss, another novelty of GENhance lies in using a cycle-consistency objective to also train the discriminator on GENhance-generated sequences. This additional training objective helps better rank and filter out the best-performing generations while the trained latent space allows us to steer the generations towards the target extrapolated distribution.
>
> - Controllability in boosting text attributes from “Neutral” to “Positive” when “Positive” samples were not in the training set:
> We wish to point out that this is precisely the objective of GENhance, i.e. to extrapolate beyond the distribution of labeled training samples. For the SST-5 experiments, the generator models were initialized with the pretrained T5 model which already has learned text representations. Through GENhance’s contrastive loss, the latent vector $z_{||}$ learns to align with the sentiment of the text, i.e. more negative (“Strongly-Negative”) samples output more negative $z_{||}$ values while more positive (“Neutral”) samples output more positive $z_{||}$ values. During the generation phase, we exploit the alignment of $z_{||}$ with the sentiment to generate samples more positive than the training data by adding perturbation to $z_{||}$ and decoding the generation with this new latent vector (Equation (9)).
> Since SST-5 samples are labeled as discrete classes of sentiment, unlike the continuous protein stability ddG values, we report the results of generated classified as “Weakly-Positive” and “Strongly-Positive” to observe how far the methods are able to extrapolate past the training data. While the main aim of the paper is to study extrapolation rather than controllability, the level of positivity of the generated samples may be controlled by how much $z_{||}$ is perturbed before the decoding phase.
>
> *About modeling:*
> - Word perturbation to smoothen latent space:
> We thank the reviewer for the discussion. In the natural language domain, a single word change may drastically change the meaning of the whole text sentence and hence, heuristics may have to be used as constraints to ensure the perturbed text is still similar to the original text. This may limit its application in other fields as such heuristics are usually domain- or application-specific. We use the WAE-MMD objective to smoothen GENhance’s latent space due to its good performance in applications with discrete sequences.
>
> - Choice of $z_{||}$ and $f_{||}$:
> We have chosen the simplest version of $z_{||}$ as a scalar and $f_{||}$ as an identity operation to have GENhance as similar as the baselines’ architectures to have a fair comparison in our experiments.
>
> *About experiments:*
> - Comparison with controllable text generation:
> Controllable text generation models such as PPLM and GeDi cannot be directly applied to our problem setting of generating samples with the target attribute beyond the training distribution, since both require discriminators to be trained on samples labeled with the target attribute. By definition, this group of samples is not available in our problem setting, as they are the distribution we are aiming to extrapolate to. To address this point, we conducted additional protein stability experiments on the CbAS and DbAS as baselines which both consider the similar problem setting. With reference to results in Table 4, the results for CbAS are: $ddG$ mean = -4.31, $PCI_{y_{\tau}}$ =  5%, $\mathbb{E}[\min]$ = -6.49 while for DbAS are: $ddG$ mean = -3.48, $PCI_{y_{\tau}}$ =  2%, $\mathbb{E}[\min]$ = -4.67. GENhance outperforms these two baselines in all three metrics, with $ddG$ mean = -7.34, $PCI_{y_{\tau}}$= 77%, $\mathbb{E}[\min]$ = -8.71.
>
> - Potential mismatch for baseline Gen-Disc:
> The two GENhance ablation variants (without the GENhance-unique objectives) in our paper are designed to consider the case mentioned by the reviewer. These ablations have a similar architecture with GENhance and also take in “Neutral”-labeled sequences during generation. GENhance’s superior performance over these baselines (Table 1 & 2) demonstrates that the designed GENhance training objectives are important for extrapolation.
>
> - Variance and performance on larger pools of generation:
> We thank the reviewer for the suggestion and will include the variance values for the other results in the revision. We wish to point out that results for larger pools of generations are available in the Appendix: Table 5 to 8 for SST-5, Table 9 to 10 for ACE. For these settings, GENhance has also generally outperformed all the baselines.
>
> - Human evaluation:
> We thank the reviewer for the suggestion. We have conducted a human evaluation study, formulated as an A/B test where three evaluators were asked to compare pairs of text in a blinded random manner to separately determine which text was more positive or more fluent than the other. The comparisons were between text generated by GENhance vs the Gen-Disc baseline, MCMC baseline, SST5 train data labeled as neutral, and SST5 train data labeled as positive. There were 100 samples evaluated for each of the 4 comparisons in the two task settings described in the paper-- which totals 800 A/B test comparisons. For each comparison, the majority answer between the three humans was assigned as the final score. The full results of the human evaluation study will be included in the updated manuscript. In short, GENhance succeeds in generating positive, fluent text that outperforms baselines. For the setting where the models were exposed to 200 positives, GENhance outperformed the Gen-Disc baseline by 68:29, outperformed the MCMC baseline by 76:17, outperformed neutral samples from the SST5 train set by 91:7, and outperformed positive samples from SST-5 train by 63:26. For the setting where the models weren’t exposed to any positives or strong positives, GENhance outperformed Gen-Disc by 53:35, outperformed MCMC by 56:34, outperformed neutral SST5 train samples by 70:18, and was comparable to positive SST5 train samples by 45:48. Likewise, the fluency of GENhance samples either match or outperform training samples and baselines.

---

> > ### Author Response · Authors · 2021-08-25
> > **awaiting response**
> >
> > Dear Reviewer, thank you again for your time and effort to produce your original review/comment. We've put considerable time and effort to address your concerns through new results (e.g. implemented additional baselines and performed a full human eval study) along with individualized responses. It'd be appreciated if you can take a look at (1) our general response and (2) our specific response to your individual review. We are eagerly awaiting your follow-up.

---

### Official Review · Reviewer_CSW1 · 2021-07-18

**Rating:** 7
**Confidence:** 3

**Summary:**

The paper considers the problem of generating sequences that have label values higher than those observed in training. An approach is proposed to encode information about the attribute to be optimized into the latent vector of an autoencoder through joint training with a discriminator and a cycle consistency objective. The model is evaluated on two different datasets, one on sentiment in movie reviews and the other on protein stability.

**Limitations And Societal Impact:**

Discussion seems reasonable and they do not overstate the ability of their approach.

**Main Review:**

The paper works on an important problem (extrapolation to unseen values of a label of interest) and proposes an interesting approach. The incorporation of the cycle consistency objective is an interesting idea and shown to be valuable in the ablations. The protein design dataset/task is an interesting contribution to the biology community. Additionally, the paper is well written with clear explanations of models+setup and descriptive usage of figures. However, there are some concerns about how it is situated relative to the literature and prior work in the area, especially wrt to claims of novelty in the task/formal setting, and possibly missing baselines/comparisons in light of this.

The formal setting here doesn’t appear to be new, other work e.g. Fannjiang and Listgarten 2020,  Brookes et al. 2019 also consider extrapolation to unseen values. The paper could be better situated relative to prior work in this area. Discussion might include Gómez-Bombarelli et al., 2018, Gupta et al., 2019, Linder et al. 2020, Fannjiang and Listgarten 2020,  and Brookes et al. 2019. That being said, the protein stability is an interesting dataset and looks useful for the broader community.

More extensive comparison to prior work seems warranted. One example being the previous work CbAS (Brookes et al. 2019). The authors write in section 2 that "Adaptive sampling (Brookes et al., 2019) uses a fixed discriminator/oracle model and iteratively learns the distribution of inputs conditioned on a desirable property using importance sampling.  We view the aforementioned techniques as complementary as GENhance proposes a model-specific architecture for optimizing attributes." Although GENhance has a model-specific architecture, there are still meaningful similarities in the motivation and problem setting of the two approaches that warrant a baseline comparison. This is further expressed by the fact that the fluorescence evaluation used by Brookes et al., 2019 is quite similar to the protein stability dataset presented in the paper. Although proper ablation studies have been performed (removing cycle-consistency and smoothing), it is ultimately difficult to judge the significance of the results in the absence of comparisons to other work in the literature. Other potential baselines include FB-VAE (Gupta & Zou 2019), GB-NO (Gómez-Bombarelli et al., 2018), RWR (Peters & Schaal, 2007), or CEM-PI (Snoek et al., 2012).

On the SST-5 experiments. Some questions around the comparison. Can the model fool the oracle by generating in a space that is adversarial for the oracle? It might be helpful to control for the fluency of generations e.g. as measured by perplexity. It looks like GENhance has significantly higher perplexity than Baseline Gen-Disc which might confound interpretation of results. Additionally, why are only 'Neutral'-labeled SST-5 sequences used as the initial sequences for the models? Why not insert Strong-negative and negative examples as well? So far, the results only show that the model learns a landscape from neutral to positive, but not necessarily negative -> neutral -> positive.

**Time Spent Reviewing:**

5

---

> ### Author Response · Authors · 2021-08-10
> **Response to Reviewer CSW1**
>
> We thank the reviewer for the thoughtful feedback. We are encouraged that you find the problem to be important, our approach interesting and the paper well-written. We also agree that our protein stability dataset should be useful to the broader community. We address your comments below:
>
> - Regarding “Formal setting doesn’t appear to be new”:
> We thank the reviewer for pointing us to references on extrapolation to unseen values of a label of interest, including Gómez-Bombarelli et al., 2018, Gupta et al., 2019, Linder et al. 2020. We have updated the related work section with discussions on these and other related papers to better contextualize prior work in this space. However, we would like to note that extrapolating labels for sequence generation models is still a fairly unexplored area, especially for sequence domains that can utilize large-scale pretraining such as NLP and proteins.
>
> - More comparison to prior work:
> As recommended, we have conducted additional experiments on the CbAS baseline. With reference to Table 4, the results for CbAS are: $ddG$ mean = -4.31, $PCI_{y_{\tau}}$ =  5%, E[MIN ]= -6.49. We observe that CbAS outperforms Baseline Gen-Disc, due to the additional finetuning step that trains the generator to sample sequences that are predicted by the discriminator to be more stable. Nonetheless, GENhance substantially outperforms CbAS in all three metrics, with $ddG$ mean = -7.34, $PCI_{y_{\tau}}$= 77%, $\mathbb{E}[\min]$ = -8.71 (Table 4). We also experimented on the DbAS baseline with results: $ddG$ mean = -3.48, $PCI_{y_{\tau}}$ =  2%, $\mathbb{E}[\min]$ = -4.67 that also underperforms GENhance. GENhance’s improvement over the baselines is contributed by two main features, as detailed in line 314 of the paper: a) latent space to steer generation towards a more stable region and b) discriminator that is trained on GENhance-generated sequences on top of the training samples.
>
> *Questions on SST-5 experiments:*
> - “Can the model fool the oracle by generating in a space that is adversarial for the oracle? It might be helpful to control for the fluency of generations.” :
> We thank the reviewer for the discussion and wish to point out that the mean perplexity value for GENhance-generated samples is comparable to the mean perplexity value of the test set for SST-5 (90.5 v 101.3 in Table 1). In addition, we have added a new human evaluation study on the model generations. With respect to fluency in A/B tests, GENhance generates text that is as fluent as samples in the train set labeled as positive (39:39) and more fluent than samples in the train set labeled as neutral (52:30). Hence, we believe that the GENhance-generated sequences have similar fluency and data distribution to the original SST-5 and feel confident about the oracle evaluation setup. Moreover, since the model does not receive any iterative feedback or gradient updates based on oracle predictions, it cannot optimize for performance on the oracle model.
>
> - Why are only 'Neutral'-labeled SST-5 sequences used as the initial sequences for the models?
> In practical applications of generating sequences that have label values higher than those observed in training, we want to push the samples with the highest attribute so far, even higher. This intuition was primarily driven by the choice of initial sequences in the continuous label space, e.g. proteins. For the SST-5 experiments for generating “weak positive” and “positive”, the samples in the training set with the highest attribute are “neutral” so they were chosen as initial sequences. To address the question about the sentiment landscape, we have performed additional experiments using “negative” samples as initial sequences for GENhance. In the setting of Table 1, GENhance can output generated sequences, with an initial “negative” sequence as input, to “positive” with 75.8% rate in the metric “% positive”, which is not a large drop as compared to the original results of 98.7%, where “neutral” sequences are also included. This demonstrates GENhance has learned the full landscape across “negative”->”neutral”->”positive”.

---

> > ### Author Response · Authors · 2021-08-25
> > **awaiting response**
> >
> > Dear Reviewer, thank you again for your time and effort to produce your original review/comment. We've put considerable time and effort to address your concerns through new results (e.g. implemented additional baselines and performed a full human eval study) along with individualized responses. It'd be appreciated if you can take a look at (1) our general response and (2) our specific response to your individual review. We are eagerly awaiting your follow-up.

---

> > > ### Comment · Reviewer_CSW1 · 2021-09-01
> > > **Re: awaiting response**
> > >
> > > Thank you for the detailed response. The new baseline significantly strengthens the work and as a result I am increasing my score.
> > >
> > > I agree with you that "extrapolating labels for sequence generation models is still a fairly unexplored area." However I think it's important for the paper to be measured in its claims of formal novelty given the history of prior work on this problem.

---

> > > > ### Author Response · Authors · 2021-09-02
> > > > **Re:**
> > > >
> > > > Thank you for the comments and increasing the score. We are happy to hear that the additional results significantly strengthen the work, and we certainly agree!
> > > >
> > > > Lastly, in our revision edits for the manuscript, we made sure to properly address all prior work noted by the reviewers and will not overclaim formal novelty. Thanks again.

---

### Official Review · Reviewer_6v4B · 2021-07-20

**Rating:** 6
**Confidence:** 5

**Summary:**

GenEnhance proposes a probabilistic sequence autoencoder combined with a discriminator for ranking sampled sequences, with the goal of enhancing capability of generating better sequences (in term of certain attribute). For this purpose, a disentangled auto encoder is used, which leverages a contrastive loss on top of  WAE-MMD loss, reconstruction loss and a cycle-consistency objective.



**Limitations And Societal Impact:**

See above.

**Main Review:**

The paper addresses an important area of controllable generation of better samples. The model formulation leverages known architectures and loss objectives. The performance of the model is shown on two different applications - positive sentence generation using a sentiment dataset and more stable protein sequence generation using a protein variant dataset.

A number of baselines has been considered including a random--MCMC, MCMC from a T5 model, as well as a number of ablation experiments have been performed. The paper is nicely and crisply written.

The paper work does not compare Genhance with natural language baselines from other published works such as PPLM or GeDi. Also, the paper does not discuss or compare with a number of recent works on controllable biological sequence generation (e.g.  Nat Biomed Eng 5, 613–623, 2021. https://doi.org/10.1038/s41551-021-00689-x, https://doi.org/10.1371/journal.pcbi.1008736) as well as the line of work that performs optimization on the latent space to generate better sequences, particularly in the context of molecular/biological sequence optimization (http://proceedings.mlr.press/v97/brookes19a.html, https://pubs.acs.org/doi/10.1021/acscentsci.7b00572, https://arxiv.org/abs/2011.01921). The paper should include this line of work in the related work section.

Also, it is not apparent from the presented results to what extent generated sequences are novel (for both English and protein tasks). The authors should include a novelty analyses of the generated samples.

For both experiments, the work considers controlling generation by a single attribute. Can authors include experiments to show how Genenhance enhances better quality sample when one needs to control two attributes.

Also, can the authors investigate the effect on Genhance performance  when there are no/less number of stable sequences  in the training data.

**Time Spent Reviewing:**

2.5 hrs

---

> ### Author Response · Authors · 2021-08-10
> **Response to Reviewer 6v4B**
>
> We thank the reviewer for the thoughtful feedback. We are encouraged that you find the problem considered in the paper to be important and the paper to be well-written. We address your comments below:
>
> - Comparison with NLP baselines such as PPLM and GeDI:
> PPLM and GeDi cannot be directly applied to our problem setting of generating samples with the target attribute beyond the training distribution, since both require discriminators to be trained on samples labeled with the target attribute. By definition, these samples are not available in our problem setting, as they are in the distribution we are aiming to extrapolate to.
>
>
> - Regarding comparison and discussion of a number of recent works on controllable biological sequence generation as well as the line of work that performs optimization on the latent space to generate better sequences:
> We thank the reviewer for pointing us to these papers for discussion. As advised by the reviewer, we have conducted additional experiments on two more baselines (CbAS and DbAS). These two baselines are implemented with the same architecture as Baseline Gen-Disc, as recommended by the other reviewer. With regards to Table 4, the results for CbAS are: $ddG$ mean=-4.31, $PCI_{y_{\tau}}$= 5%, $\mathbb{E}[\min]$=-6.49 while for DbAS are: $ddG$ mean=-3.48, $PCI_{y_{\tau}}$= 2%, $\mathbb{E}[\min]$=-4.67. Even though the CbAS fare better than the simpler baseline Gen-Disc setup, we observe that GENhance outperforms CbAS on all the 3 metrics measured.
> Here, we further discuss the works mentioned by the reviewer and will include the following in the revised version.  CbAS and DbAS both adaptively sample training samples for the generator model using the discriminator model’s score to upweight samples predicted to be better-performing (more stable). CbAS is an improved version of DbAS which also re-weights samples based on how close they are to the original training data. Das et al. 2021 train VAE to learn latent space and use latent space classifiers to sample latent vectors through rejection sampling and decode them into sequences that would have the target attribute/label. Hawkins-Hooker et al. 2021 also decode generations of a VAE by conditioning on latent vectors that correspond to the target attribute/label. Hoffman et al. 2020 seek to optimize molecular designs by using zeroth-order optimization on query-based prediction of candidate molecules’ properties. Gomez-Bombarelli et al. 2018 build a Gaussian Process (GP) regression model trained with latent vectors to predict their inputs’ labels and use gradient-based optimization on the GP to find sequences with target attributes. Compared with these previous works, the core difference in our approach is the combination of cycle-consistency and contrastive discriminatory objective to train the generator and discriminator as one model.
>
>
> - Regarding “it is not apparent from the presented results to what extent generated sequences are novel”:
> We would like to point out that our main goal is to generate samples with the target attribute beyond the training distribution, not necessarily samples that are novel in the sequence space. Indeed, the mean perplexity value for GENhance-generated samples is comparable to the mean perplexity value of the test set for SST-5 (90.5 v 101.3), i.e. their degree of novelty from the train set is similar to that of the test set from the train set. However, our results on the target attribute demonstrate that GENhance can extrapolate in the label-space with respect to implemented baselines through the metrics defined in our paper.
> During the response period, we performed a human evaluation study to judge GENhance’s ability to generate text that is more positive and more fluent in an A/B test setup described in the shared response above. The following results are for the question “Which text is more positive than the other?”. For the setting where the models were exposed to 200 positives, GENhance outperformed the Gen-Disc baseline by 68:29, outperformed the MCMC baseline by 76:17, outperformed neutral samples from the SST5 train set by 91:7, and outperformed positive samples from SST-5 train by 63:26. For the setting where the models weren’t exposed to any positives or strong positives, GENhance outperformed Gen-Disc by 53:35, outperformed MCMC by 56:34, outperformed neutral SST5 train samples by 70:18, and was comparable to positive SST5 train samples by 45:48. Likewise, the fluency of GENhance samples either match or outperform training samples and baselines.
>
>
> - Exploring two-attribute generation and investigating the effect on GENhance's performance when there is no/less number of stable sequences in the training data:
> We thank the reviewer for good suggestions on future work about controlling two attributes & more few/zero-shot settings. In the paper, we conducted experiments on the few/zero-shot setting for SST-5, where there is no/less number of positive samples in the training data. On the protein stability problem, since the stability label exists on a continuous scale, our current experiments in Table 4 represent a zero-shot setting for generating highly stable sequences not present in the training set. However, exploring a problem setting where very few training sequences are present is indeed an important direction for future research. In sum, future work in these directions should be a considerable contribution to the field.

---

> > ### Author Response · Authors · 2021-08-25
> > **awaiting response**
> >
> > Dear Reviewer, thank you again for your time and effort to produce your original review/comment. We've put considerable time and effort to address your concerns through new results (e.g. implemented 2 additional baselines and performed a full human eval study) along with individualized responses. It'd be appreciated if you can take a look at (1) our general response and (2) our specific response to your individual review. We are eagerly awaiting your follow-up.

---

### Author Response · Authors · 2021-08-10
**Overall Response to All Reviewers**

We would like to thank all reviewers for the thoughtful comments and feedback. We have addressed each reviewer’s comments separately. The following addresses common themes and feedback across all reviewers:

**GENhance outperforms two added baseline implementations for model-based optimization.** We have added two additional baselines from existing methods for model-based optimization, namely DbAS and CbAS, following the implementation recommended by one of the reviewers. The results for CbAS are: $ddG$ mean = -4.31, $PCI_{y_{\tau}}$ =  5%, $\mathbb{E}[\min]$ = -6.49 while for DbAS are: $ddG$ mean = -3.48, $PCI_{y_{\tau}}$ =  2%, $\mathbb{E}[\min]$ = -4.67. As reported in Table 4, GENhance outperforms both baselines with a $ddG$ mean = -7.34, $PCI_{y_{\tau}}$ =  77%, $\mathbb{E}[\min]$ = -8.71.

**Existing controllable generation techniques are not suited for our formulated task in extrapolation.** Some reviewers suggested a comparison with controllable generation methods in NLP, such as CTRL, PPLM, or GeDi. These methods cannot be directly applied to our problem setting as they require the target attribute to be observed in training. Our extrapolation task, by definition, involves generating sequences with attributes not seen in training.

**We have added a human evaluation study on generations. Results demonstrate GENhance samples are more positive than baselines for SST-5 experiments.** To further evaluate generated text aside from the oracle discriminator scores, we performed a human evaluation study to examine the positiveness and fluency of our text generations. The study was formulated as an A/B test where three evaluators were asked to compare pairs of text in a blinded random manner to separately determine which text was more positive or more fluent than the other. The comparisons were between text generated by GENhance vs 1) the Gen-Disc baseline, 2) MCMC baseline, 3) SST5 train data labeled as neutral, and 4) SST5 train data labeled as positive. There were 100 samples evaluated for each of the 4 comparisons in the two task settings described in the paper-- which totals 800 A/B test comparisons. For each comparison, the majority answer between the three humans was assigned as the final score. The full results of the human evaluation study will be included in the updated manuscript.
In summary, GENhance succeeds in generating positive, fluent text that outperforms baselines. For the setting where the models were exposed to 200 positives (200-Pos setup), GENhance outperformed the Gen-Disc baseline by 68:29, outperformed the MCMC baseline by 76:17, outperformed neutral samples from the SST5 train set by 91:7, and outperformed positive samples from SST-5 train by 63:26. For the setting where the models weren’t exposed to any positives or strong positives (No-Pos), GENhance outperformed Gen-Disc by 53:35, outperformed MCMC by 56:34, outperformed neutral SST5 train samples by 70:18, and was comparable to positive SST5 train samples by 45:48. Likewise, the fluency of GENhance samples either match or outperform training samples and baselines.

**We have expanded the discussion of prior literature and citations.** To better contextualize the works mentioned by reviewers, we have better contextualized how our task and model fits within existing research through an expanded Related Work section and throughout the revised manuscript (detailed discussions are in individual responses to the reviewers).

---

> ### Author Response · Authors · 2021-08-20
> **awaiting reviewer responses**
>
> Reviewers-- Again, thank you for your initial reviews. We've put considerable time and effort to address your concerns through new results and individualized responses. It'd be appreciated if you can take a look at (1) our general response and (2) our specific response to your individual reviews. Please scroll down to see your individualized response. We are awaiting your follow-up comment.

---

### Decision · Program_Chairs · 2021-09-27

**Decision:**

Accept (Poster)

**Comment:**

The paper presents a method for generating sequences with attributes or characteristics that are not present in the training data. It is based on an encoder-decoder  model, trained according to a regularized loss. The applications concern two domains: more positive sentence generation and more stable protein sequence generation. The reviewers agree that the topic, extrapolating to unseen attribute values for sequence generation is important and that the technical contribution – through the combination of a contrastive loss and a cycle consistency loss is original.

The initial reviews highlighted weaknesses concerning two main aspects: the positioning w.r.t. the literature and the lack of experimental comparisons with SOTA baselines.  The authors provided a strong rebuttal answering with details to the different questions and remarks. They provided comparisons with two new baselines suggested by the reviewers and performed a qualitative human evaluation also suggested in the reviews. They additionally introduced new ablation analyses. The reviewers have appreciated the quality of the answers and in light of this, they all raised their scores. I propose an accept.